# The Effect of the KDL Active School Plan on Children and Adolescents’ Physical Fitness in China

**DOI:** 10.3390/ijerph192013286

**Published:** 2022-10-14

**Authors:** Tiantian Tong, Xiaozan Wang, Feng Zhai, Xingying Li

**Affiliations:** 1College of Sports, China University of Mining and Technology, Xuzhou 221000, China; 2College of Physical Education and Health, East China Normal University, Shanghai 200241, China

**Keywords:** KDL Active School Plan, different level students, physical fitness, teaching intervention, empirical study

## Abstract

The development of physical fitness among Chinese children and adolescents is not fundamentally improving, and an exploration of effective ways to promote it is an urgent need. Research into physical fitness promotion in schools is increasingly deepening worldwide. However, the implementation and verification of intervention programs with local characteristics in accordance with China’s national conditions are relatively weak. This study conducted a randomized controlled trial to examine the effects of the KDL (Know it, Do it, Love it) Active School Plan (KDL-ASP) on children and adolescents’ physical fitness. A total of 596 students from level two (2nd-grade students) to five (11th-grade students) in China were assessed in terms of their physical fitness. Of these, 308 students were randomly selected to participate in the KDL-ASP, which uses a combination of indoor and outdoor sports activities in which teachers, parents, and students participate together. The remaining 288 students performed conventional physical activities. After one school year of intervention with the KDL-ASP, the physical fitness of the children and adolescents improved. The improvements in the speed of level two girls, the strength of level four boys, and the lung capacity of level five boys were the most obvious. These results demonstrate the viability of indigenized intervention in schools to improve physical fitness and suggest that KDL-ASP needs to be considered throughout the whole progress of physical education learning for children and adolescents.

## 1. Introduction

Physical fitness is an important health indicator in children and adolescents [1], and research shows that physical fitness in childhood and adolescence can influence fitness in later adulthood [2]. Stanley Hui found that adolescents who followed PA guidelines were more likely to be in the health fitness zone of aerobic and muscular fitness [3]. In fact, 81% of children and adolescents are insufficiently physically active globally [4]. Physical inactivity is a serious threat to the health and wellbeing of the population [5]. The attenuation and prevention of childhood obesity is important and necessitates innovative and effective interventions to promote children and adolescent’s physical fitness.

According to the Healthy China Action (2019–2030) document, we need to strengthen physical education and extracurricular exercise and ensure that primary and secondary school students have at least one hour of physical activity every day when they are in school. By 2030, 60% or more of students will meet the national physical fitness standards [6]. The physical fitness of children and adolescents in China is a key factor in achieving a “Healthy China”; however, the overall level of physical fitness is still dropping [7]. This is due to insufficient physical education courses being mandated by the state and the low quality of teaching. The poor form and content of sports between classes have led to a lack of activity among children and adolescents [8]. Therefore, we need to urgently explore paths towards and the effects of promoting the healthy development of children and adolescents.

In order to improve the level of physical activity and physical fitness, countries around the world have developed and implemented different effective plans, such as PESS (Physical Education and School Sport) in Britain [9], CSPAP (Comprehensive School Physical Activity Program) in the United States [10], etc. However, physical fitness promotion is affected by national, political, economic, cultural, and other factors, and thus a localized governance path is more effective. Having learned from advanced foreign experiences, our country has developed and promulgated “Chinese Physical Education Curriculum Standards” and the localized Chinese Healthy Physical Education Curriculum Model. KDL sports and health courses with Chinese characteristics were launched in 2018 to help PE teachers to deliver high-quality teaching. KDL aims to help students master structured sports, health knowledge, and skills in an active atmosphere, improve their motor abilities (Know it), use them actively in daily life, develop healthy behavior (Do it), and eventually form sports hobbies and specialties, cultivate sports morality, and benefit from this for life (Love it). KDL tries to cultivate students who “understand (sports) culture”, “have (sports) ability”, and “have (sports) enthusiasm” [11]. Research has shown that the implementation of the KDL PE curriculum has significantly improved the development of gross motor skills in children aged 5–6 [12], effectively promoted the development of the overall level of physical health of children aged 6–9 [13], and significantly improved the speed, strength, endurance, and flexibility of students [14].

Based on the physical education and health curriculum concept of KDL and foreign advanced experience, the KDL Active School Plan (KDL-ASP) was formed under the support of social ecology theory, the transtheoretical model, and the life course health development model framework. Further theoretical analyses of KDL-ASP are necessary to help us to understand the factors that improve the physical fitness of children and adolescents from many perspectives so that we can highlight its scientific and rational nature. (1) Social ecology theory. This theory emphasizes that individual learning behavior is formed via interactions with multi-dimensional environment systems and are especially influenced by social, natural, and internal environments [15]. KDL-ASP pays close attention to fully exploring the connections between individuals, people, organizations and other systems to create better supportive environments for the promotion of physical fitness. (2) Transtheoretical model. This theory synthesizes various theories to research individual behavior changes systematically. The core of this theory stipulates that the health behavior change process is continuous and related, involving the precontemplation stage, contemplation stage, preparation stage, action stage, and maintenance stage [16]. KDL-ASP attaches importance to various positive factors that promote individual health behaviors and integrates them with various ways to help master a healthy lifestyle. (3) Life-course healthy development model. This theory focuses on understanding healthy development from the perspectives of “processes”, “relationships”, and “systems”, emphasizing the characteristics of unfolding, complexity, timing, plasticity, adequacy, and harmony [17]. KDL-ASP accommodates local social and cultural changes, attaches importance to policy advantages, and plays a guiding role in schools and families to realize the integration of the social ecological environment for healthy development and sports activities inside and outside the school.

In summary, KDL-ASP focuses on child and adolescent health literacy cultivation and embraces the fusion education concept for instruction, which ensures that children and adolescents participate in MVPA for at least 60 min every day in school through comprehensive sports activities that take place in the classroom and outside to improve their physical and mental health. The plan covers five elements: active classroom, active playground, energetic teachers, energetic students, and dynamic environment (see Figure 1). KDL-ASP is the latest achievement in sports and health course reform and innovation in China. Discussing its influence on child and adolescent physical fitness can provide the practical basis for a wider range of applications in the future.

The purpose of this study is to analyze the positive changes to the physical fitness indexes of children and adolescents after intervention in the form of the implementation of the KDL-ASP. We hypothesized that KDL-ASP would significantly improve the physical fitness of children and adolescents at different ages and for both genders.

## 2. Methods

### 2.1. Participants

Based on the different levels of regional economic development in China, the development of physical education degree will exhibit certain differences. This study randomly selected economically developed areas in Shandong and Jiangsu provinces in eastern China and an economically backward area in Xinjiang autonomous region in western China. The three schools have two separate campuses (the software and hardware facilities are similar, and the students were from similar backgrounds), one is the intervention campus and the other is the control campus. The three schools cover primary school, junior high school, and high school, which encompass the second, third, fourth and fifth levels of physical education in China (i.e., grades 3, 5, 7, 8, 9, and 11). Children with major diseases and abnormal physical development who could not participate in physical activities were excluded, and a total of 596 children and adolescents were included. According to the design of the physical education and health curriculum standards, the learning stages are divided into six levels. Levels one to five are equivalent to grades 1–2, 3–4, 5–6, 7–9, and high-school students, respectively [18]. A total of 308 students were included in the intervention group (IG), and 288 students were included in the control group (CG). Detailed information on both groups is shown in Table 1. All subjects understood the study, signed informed consent forms, and were able to actively participate in the intervention. The experiment met the ethical requirements and was approved by the Human Experiment Ethics Committee of East China Normal University (approval number: HR222-2019).

### 2.2. Measures

Test items and equipment. The test items followed the National Physical Fitness Standards for Students (NPFSS), which includes height, weight, lung capacity, 50 m running, sit and reach, 1 min sit-ups/pull-ups, 50 × 8 shuttle running, and 1000/800 m [19]. The testers were professional PE teachers, and the test methods, equipment, and operation requirements followed the relevant NPFSS documents [20] (see Table 2).

### 2.3. Intervention Procedures

Intervention time. The intervention period was from September 2019 to June 2020, but holidays and COVID-19 disrupted offline teaching for about three months in total. During the intervention period, IG students had at least 60 min of MVPA every day, including physical education and health classes.

Intervention design. The children and adolescents participating in the study were divided into two groups (IG and CG) according to the requirements of the experiment. The experimental intervention methods were different, but the teaching level of the PE teachers, the physical education learning environment of the students, and the students’ levels and grades were the same. IG students participated in the KDL-ASP, while CG students carried out regular school sports activities, such as regular PE classes, recess, and sunshine sports activities based on athletics and ball games. These regular physical activities lasted the same amount of time in both the IG and CG. In terms of the main factors affecting physical fitness, such as exercise frequency, exercise intensity, duration, and exercise content [21], the IG followed the specific contents of the KDL-ASP (see Table 3). The specific requirements are shown in Table 4. PE teachers monitored the exercise intensity by having some students wear heart-rate bands during the intervention. The specific intervention process is shown in Figure 2.

In order to reduce the impact of time intervals and climate in each experimental school on the results of physical fitness tests, the PE teachers organized a pre-test for the IG and CG students in late September. In late May of the following year, a post-test was carried out for all students in the same venues with the same testers and methods.

### 2.4. Teaching Training

Based on a project by the National Social Science Foundation of China (NSFC), we formed a research community of interventional school PE teachers and university experts who actively participate in sub-projects. Under the leadership of the general project group (GPG), and on the basis of the previous completion of national recruitment interventional school projects, the team of graduate students in the same research direction as myself organized the training of the KDL-ASP plan in July 2019. Based on the comprehensive consideration of the region, school scale, school physical education hardware and software facilities, and the enthusiasm of the PE teachers, forty-six full-time PE teachers and four principals in the interventional schools from four public primary and secondary schools were selected to participate in the intervention. The principal of each interventional school committed and ensured the complete participation of the school in the intervention activities. We conducted three days of intensive training, in which the theory and practice were combined. The manual “KDL-ASP Action Guide” was uniformly distributed to the trained PE teachers so that they could systematically learn about the concepts and main contents of the KDL-ASP framework, understand the implementation cases, clarify the specific application principles, put forward implementation plans, and master the use of the monitoring equipment. Three KDL classes were conducted by the training team in the teaching practice and seminars (primary school, junior high school, high school), and then the trainees simulated, designed, and showed the different levels of KDL teaching practice. The training team provided feedback on the problems and deficiencies. During the training, a WeChat group was established to continuously collect the questions and thoughts of trainees. The training team provided guidance and answered questions online. During one school year of intervention, the trainees were assisted with integrated physical activity resources and learning modules on the KDL Physical Education and Health Curriculum network platform. Supervision seminars were organized online by the training team once a month. The length of students’ campus high intensity sports activities was reported every day, the number of school PE classes was reported every week, the type of students who participated in extracurricular sports activities was reported every day, the number of brain break-ins conducted in the literacy class was reported weekly, and the proportion of school faculty who participated in physical activities and the frequency of parent–child physical activities were reported monthly. Sport cultural activities were conducted each semester in the interventional school and were monitored via video, photos, data, and other materials during the implementation to monitor and evaluate the fidelity of the intervention program implemented by the interventional school.

### 2.5. Statistical Analysis

This study had three independent variables (group, sex, and level) and one dependent variable (physical fitness achievement). The scores of each test index of physical fitness were converted into a percentage according to the score of the National Physical Fitness Standards for Students (NPFSS). Table 5 present the total score calculated by the weight coefficient of the physical fitness index determined in the NPFSS [20]. The researchers entered the data and generated the results using IBM SPSS 23.0. Firstly, multi-factor analysis of variance was used to explore the influence degree of different independent variables on the progress of total physical fitness scores. Secondly, an independent sample t-test was used to explore the differences in physical fitness test indexes’ progress between the IG and CG at each level and between boys and girls in the IG. Lastly, a one-way ANOVA was used to explore the difference in the progress of physical fitness test indexes between boys and girls at different levels. When comparing the differences in independent sample means, we used Cohen standards to judge effect sizes (ES): 0.2 (small), 0.5 (medium), and 0.8 (large) [22]. ES were generated and η^2^ magnitudes of 0.01, 0.06, and 0.14 were interpreted as small, medium, and large effects, respectively, in the one-way ANOVA [23].

## 3. Results

### 3.1. The Impact of Physical Fitness Total Scores Improvement for Children and Adolescents

Due to the different levels of the pre-test of physical fitness indexes between the IG and CG, this study focused on the progress. The multi-factor analysis of variance showed that the total score improvement of the physical fitness index showed the effects of intervention and gender and interactions between intervention and level (F _intervention_ = 12.029, *p* < 0.01; F _sex_ = 4.913, *p* < 0.05; F _level_
_×_
_intervention_ = 9.629, *p* < 0.01) (see Table 6). Figure 3 presents the simple effect analysis of the interaction between the intervention and the level, which shows that after the intervention trial, regardless of the level, the average improvement in the total physical fitness scores of children and adolescents participating in the intervention was higher than that of the students who did not. Therefore, the difference of the intervention will be analyzed separately from the aspects of gender and level. Due to the excessive number of results, each section only presents and analyzes the results with significant differences.

### 3.2. Different Levels and Genders’ Progress in Some Indexes between IG and CG

Since the group consisted of two variables, an independent sample t-test was used. Table 7 shows that there were significant differences in students’ progress in terms of the speed and total physical fitness scores for girls at level two (*p* _level 2 girls’ speed_ < 0.05; *p* _level 2 girls’ total_ < 0.05, d = 0.764–0.782). There was a significant difference in the improvement of endurance among girls at level three (*p* _level 3 girls’ endurance_ < 0.05, d = 0.637). There were significant differences in the strength, endurance, and total physical fitness scores of boys at level four (*p* _level 4 boys’ strength_ < 0.01, d = 0.675; *p* _level 4 boys’ endurance_ < 0.05, d = 0.412; *p* _level 4 boys’ total_ < 0.05, d = 0.377), among which the difference in strength was the most significant. There were very significant differences in terms of the flexibility, strength, and physical fitness progress of girls at level four (*p* _level 4 girls’ flexibility_ < 0.01; *p* _level 4 girls’ strength_ < 0.01; *p* _level 4 girls’ total_ < 0.01, d = 0.565–0.811). In addition, the mean improvement of level two, three, and four students who underwent IG was higher than that of CG students.

### 3.3. The Progress of Physical Fitness Indexes for Boys and Girls at Different Levels

Since the level consisted of four variables, a one-way ANOVA was used. Table 8 presents the significant differences in improvement of lung capacity, strength, and endurance in boys at various levels (*p*
_lung capacity_ < 0.01, *p* _strength_ < 0.05, *p* _endurance_ < 0.05; η^2^ = 0.061–0.212). There were very significant differences in the improvement of lung capacity and speed quality of girls at various levels (*p*
_lung capacity_ < 0.01, *p* _speed_ < 0.01; η^2^ = 0.130–0.152). Judging by the progress mean, compared with other level students, the lung capacity and speed quality of level two girls, the strength and endurance quality of level four boys, and the lung capacity of level five boys showed the greatest improvement.

### 3.4. The Progress of Physical Fitness Indexes of Children and Adolescents between Different Gender

Since gender consisted of two variables, an independent sample t-test was used. Figure 4 presents the significant differences in the improvement of strength between boys and girls (*p* _strength_ < 0.01) in the IG. Judging from the mean, the improvement in the strength of boys was significantly higher than that of girls. With the exception of flexibility, the improvement in all physical fitness index scores and total scores of boys was significantly better than that of girls.

## 4. Discussion

The above results have shown that both intervention and gender impact physical fitness. The impact on children and adolescents varied at different levels. Apart from level five, some of the physical fitness indexes of IG students at different levels were significantly different from those of CG, including speed and total physical fitness in level 2 girls; endurance and strength in level 3 girls; endurance, flexibility, and total physical fitness in level 4 boys; and strength and total physical fitness in level 4 girls (see Table 7). Studies have shown that having multiple types of in-school PA is an important factor for improving physical fitness. Relying only on physical education and health courses to continuously improve physical fitness without a global physical education concept would be unrealistic [24]. Increasing the quantity and quality of PE, integrating it into health education classes, changing the school environment, and increasing after-school PA can effectively promote adolescent participation [25]. Sport and fitness atmosphere, sport media, and children’s physical activities are positively correlated [26]. The comprehensive intervention of PA based on home-school linkage has a more obvious positive effect on the physical fitness of primary school students [27]. KDL-ASP integrates the “big sports concept” into the daily life of children and adolescents, breaks the situation fundamentally via school alone, focuses on the school as the center, integrates the resources of PA in and out of school, and involves students, school staff, and parents, etc., to form a collaborative linkage community, create a positive and supportive sport environment, and increase the time and opportunity for children and adolescents to participate in PA in school [28]. KDL-ASP increased the progress of level 2, 3, and 4 students in terms of speed, strength, and physical fitness more so than regular PE lessons. However, the physical fitness of level five students conspicuously did not change, which may be due to the greater academic pressure on high school students. This may reduce their enthusiasm to participate in PA and lead to excellent rates of physical fitness for high school students, which have been significantly lower than those of junior high schools over the past thirty years in China [29].

In addition, from the perspective of the physical fitness indexes of children and adolescents at different levels, there were significant differences in terms of progress in speed, strength, and endurance. The reason is that in the process of growth and development of children, the various forms and functions suitable for a special stage are unbalanced and gradually develop, so there is a “sensitive period for the development of physical fitness” [30]. Professor Mao pointed out that China is facing the dilemma of inaccurate analysis in terms of enhancing students’ physical fitness, so we should pay attention to the exercise prescription for students’ physical sensitivity period to improve physical fitness efficiently [31]. Children who ran 50 m in their sensitive period frequently saw their speed grow faster, which generally increased most rapidly in 9–13-year-olds [32]. Their strength and endurance gradually grow with age, and the critical period of development is 12–15 years of age [33], which is also the period in which flexibility increases, particularly in 14–15-year-olds [34]. The sensitive period of the strength and speed of boys is later than that of girls [35]. After the KDL-ASP, the speed of level two girls and the strength of level four boys improved significantly, and the progress in these domains were the largest of all levels, which also confirms that KDL-ASP intervention could have a positive effect during their sensitive periods. Our analysis found that the ES of level four boys’ endurance and total physical fitness improvement difference between IG and CG was low and showed only a small difference. This may be related to the implementation of the physical examination policy in junior high school in China to assure that they improve their physical fitness; however, it is still a kind of exam-oriented education, which is not conducive to having children and adolescents continue to participate in physical activities and develop the habit of physical exercise [36].

In terms of lung capacity, IG showed significant differences between the various levels of boys and girls, the improvement between girls at level two and boys at level five was greater than that of the same sex students at other levels. On the one hand, through intervention, the respiratory rate of children is increased to maintain the body’s demand for oxygen during exercise, thereby improving the ventilatory function and strengthening respiratory muscle function [37]. On the other hand, lung capacity has the characteristic of continuously increasing with age [38], with lung capacity increasing more through intervention. Previous studies have revealed that comprehensive physical activities, which contain high-quality PE, two-exercise innovation activities, sports clubs, and parent–child activities, can effectively improve students’ participation in sports [39]. We believe that in addition to the influence of the sensitive period, the KDL-ASP intervention can make students participate more, improve their physical fitness, and lung capacity.

Judging from the intervention effects of the total physical fitness scores, the KDL-ASP improved the progress of boys and girls to a certain extent. The improvement in the total physical fitness scores of boys is better than that of girls. There are significant differences between the two genders’ psychological needs, motivation, and satisfaction among middle school students when participating in leisure sports, and the correlation of boys was higher than that of girls [40]. A survey on PA among children and adolescents in China found that boys’ PA in and out of school was significantly better than that of girls [41]. Therefore, the intensity of exercise in boys may be greater, and the overall level of physical fitness is significantly higher. In fact, at a lower age, girls are better at running and jumping than boys [42]. Coordination and balance are also more advanced among younger girls, and these advantages in motor skills could be positive factors for participation in sports, but with advancing age, these advantages of girls gradually weaken, especially strength and endurance. The negative attitude of girls towards participation in PA is caused by conservative parenting environments in their families. The healthy development model of life course believes that there are three important stages in the healthy development of individual life processes, namely incubation period, cumulative period, and critical period. Each stage has the potential to increase the risk of adverse health outcomes later in life [43]. That is to say, the long-term improvement of the physical fitness level of children and adolescents does not only rely on a certain stage but also needs to trace back to the source. The KDL-ASP intervention in preschoolers in the incubation stage pays more attention to the girls’ physical healthy development to improve the school and family’s environment for girls’ participation in sports. While focusing on the critical period of children’s physical healthy development, we can use a variety of forms and contents of extracurricular physical activities throughout the whole physical education learning period to stimulate students’ participation in sports, accumulate more positive health effects, and gradually form good physical exercise habits.

This study has two limitations. (a) This study only focuses on the progress of some skill-related physical fitness indexes. Future research should try to add health-related physical fitness indexes to comprehensively evaluate the physical health level of children and adolescents in China. (b) The sample size in this survey is not enough to represent the full effect of the KDL-ASP intervention. In the future, it is necessary to expand the scope of the intervention and increase the number of students to include more children and adolescents.

## 5. Conclusions

The implementation of KDL-ASP for one school year can significantly improve the physical fitness of children and adolescents, including their strength, endurance, speed, and lung capacity, and its influence on the physical fitness of boys is greater than its influence on girls. These research result showed that the KDL-ASP, which comprises active classroom, active playground, energetic teachers, energetic students, and dynamic environment, was feasible for school physical education, which provides a practical reference for promoting the healthy development of children and adolescents. In the future, the long-term effects of the program should be monitored in a larger population with more indexes.

## Figures and Tables

**Figure 1 ijerph-19-13286-f001:**
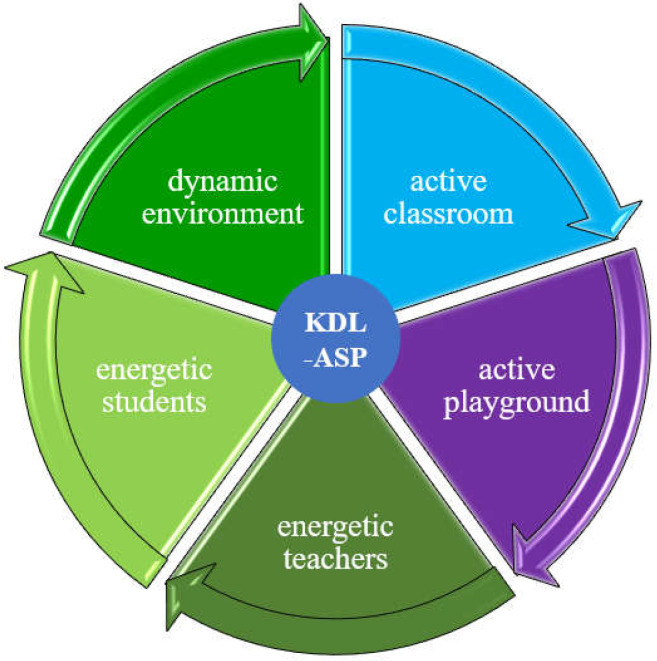
Elements of KDL-ASP.

**Figure 2 ijerph-19-13286-f002:**
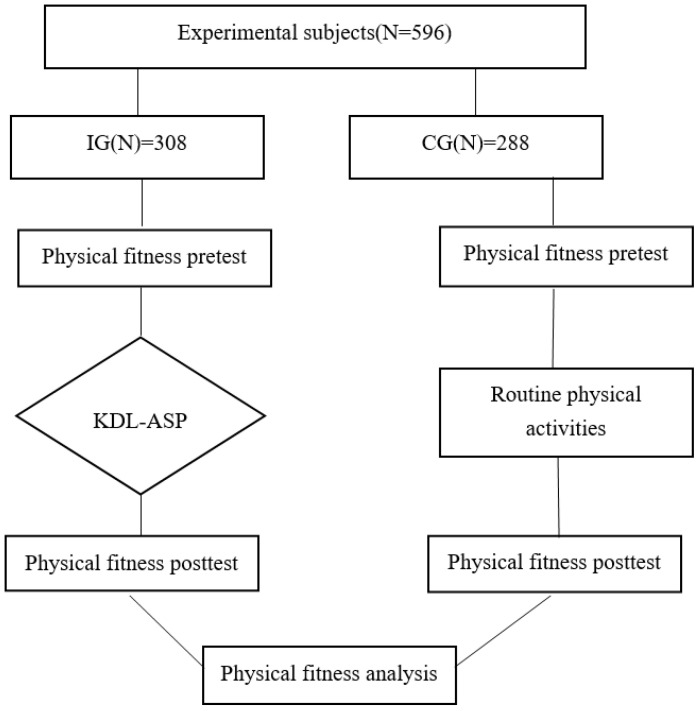
Flow chart of the experiment.

**Figure 3 ijerph-19-13286-f003:**
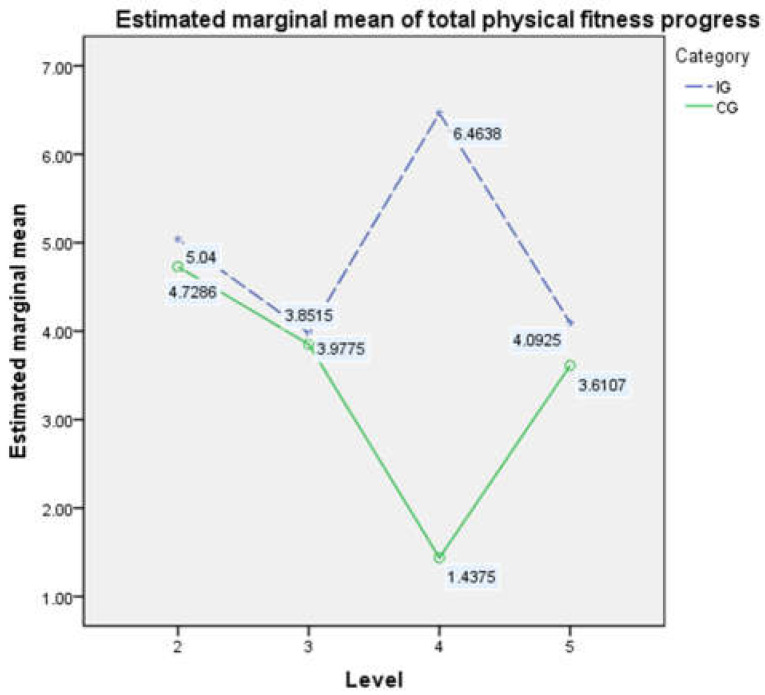
The interaction effect between the intervention and the level of progress of physical fitness scores after the experiment.

**Figure 4 ijerph-19-13286-f004:**
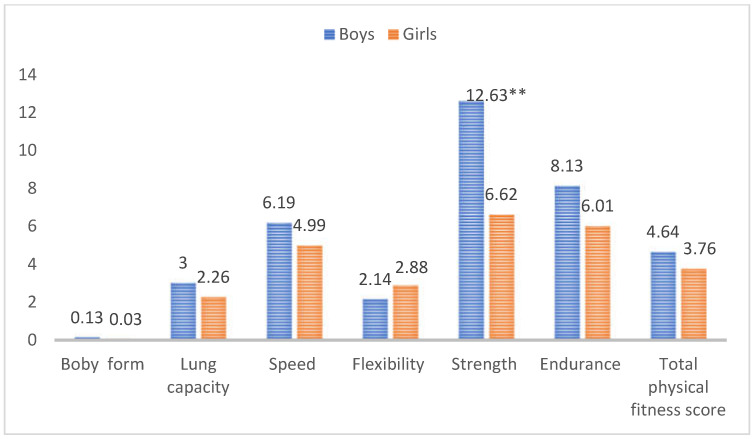
The independent-samples t test of physical fitness scores progress between different gender. Note: ** means at the alpha 0.01 level.

**Table 1 ijerph-19-13286-t001:** List of gender distribution of research subjects.

Level	Grade	Gender	IG	CG	Total	Percentage
Level 2	3rd grade	Boys	20	15	74	12.4%
		Girls	20	19		
Level 3	5th grade	Boys	24	25	91	15.3%
		Girls	22	20		
Level 4	7th grade	Boys	19	22	82	13.8%
		Girls	23	18		
	8th grade	Boys	25	20	91	15.3%
		Girls	25	21		
	9th grade	Boys	31	26	108	18.1%
		Girls	25	26		
Level 5	11th grade	Boys	40	44	150	25.1%
		Girls	34	32		
	Total	Boys	159	152	596	100%
		Girls	149	136		

**Table 2 ijerph-19-13286-t002:** List of test items and physical fitness experiments.

Physical Fitness	Test Items	Test Equipment
Body form	Body Mass Index ^abc^	Height and weight instruments
Lung capacity	Lung capacity (mL) ^abc^	Automatic Electronic Spirometer
Speed	50 m running (s) ^abc^	Stopwatch; Starting flag; Whistle
Flexibility	Sit and reach (cm) ^abc^	Sit and reach tester
Strength	1 min sit-ups (number) ^abc^	Stopwatch; Exercise mat
	1 min pull-ups (number) ^c^	Horizontal bar
Endurance	50 × 8 shuttle running (m/s) ^b^	Stopwatch; Starting flag; Whistle
	1000/800 m (m/s) ^c^	Stopwatch; Starting flag

^a,b,c^ stand for 3rd grade, 5th grade, and 7–11th grade, respectively.

**Table 3 ijerph-19-13286-t003:** The specific contents of the KDL-ASP.

Active Factor	Exercise Contents	Specific Modalities
Active classroom	Brain break in academic subjects	Invent or imitate video movements in Chinese, mathematics, and foreign language subjects every day for 3–5 min involving low- to moderate-intensity physical activities.
Active playground	KDL PE curriculum	The process in a physical education class was: (1) 5 min of warm-up with a fun game. (2) 20 min of combined skill practice involving setting up a situation involving a complete skill and tactics, practice, and competition. (3) 10 min of physical exercise that complements the learned skills and interesting content. (4) 5 min of relaxation with a game.
	Active aerobic exercise	Create rhythmic exercises with the characteristics of their school and promote them in recess activities to improve students’ physical fitness comprehensively.
	Campus Guinness challenge	According to the students’ level, PE teachers determine the Guinness records of students at different ages and organize a “challenge qualification competition” involving rope skipping, shuttlecock kicking, and other sports. The selected students will be eligible to challenge for the record. The “3 + 1” cycle (three weeks of practice, one week of competition) rotation mode is adopted. Students will constantly refresh their personal best record after a competition. At the end of the semester, the students with the highest records in each sport are awarded accordingly.
	Sport group challenge	Play basketball, football, table tennis, martial arts, and other sports that the students choose according to their interests for 30 min every week. At the end of each month, there is a peer challenge in the class. At the end of each semester, organize a sport challenge in the grade, and according to the results of the challenge promote the students’ health.
Energetic teachers	Walking outdoors with teachers	According to the different grades of the teachers, the school regularly organizes outdoor hiking activities for teachers and staff, and the hiking distance is 5 km each time.
Energetic students	Parent–child carnival	With the school as the venue, students of the same grade and their parents are recruited to participate in the sports carnival every month. The content of the activities is primarily fun track and field games, and the winners will be given material rewards at the end of the activity.
Dynamic environment	Sport and health promotion campaign	Post periodically on the class bulletin board and propaganda board about physical health.

**Table 4 ijerph-19-13286-t004:** The specific requirements of the KDL-ASP intervention.

Exercise Frequency	Duration	Exercise Contents	Exercise Intensity (HRmean)
1 time/day	20 min × 1	Actively aerobic exercise	(220–age) × (60–80%)
4 times/week (elementary school) 3 times/week (junior high school and high school)	40 min × 4 40 min × 3	KDL curriculum	125–140 times/minute 140–160 times/minute
1 time/day	3–5 min × 1	Brain break in academic subjects	100–130 times/minute
3 times/week	30 min × 3	Campus Guinness challenge; Sport group challenge	Over 160 times/minute
1 time/week	30 min × 1	Walking outdoors with teachers	(220–age) × (60–80%)
1 time/month	60 min × 1	Parent–child carnival	(220–age) × (60–80%)

**Table 5 ijerph-19-13286-t005:** The weight calculation table of the total physical fitness scores among students at four levels.

Level	Physical Fitness Weight Calculation Formula
Level 2	Standard weight for height × 0.2 + sit-ups × 0.4 + 50 m running × 0.4
Level 3	Standard weight for height × 0.1 + lung capacity × 0.2 + 50 × 8 shuttle running × 0.3 + sit-ups × 0.2 + 50 m running × 0.2
Levels 4 and 5	Standard weight for height × 0.1+ lung capacity × 0.2 + 1000/800 m × 0.3 + sit-ups/pull-ups × 0.2 + 50 m running × 0.2

**Table 6 ijerph-19-13286-t006:** The intersubjective effect test of the progress of children and adolescents’ physical fitness scores after the experiment.

Source of Variation	Type Sum of Squares	Degree of Freedom	MS	F	*p*
Correct model	1725.462 ^a^	15	115.031	4.933	0.000
Intercept	7515.129	1	7515.129	322.312	0.000
Level	41.192	3	13.731	0.589	0.622
Intervention	280.481	1	280.481	12.029	0.001 **
Sex	114.563	1	114.563	4.913	0.027 *
Level × Intervention	673.511	3	224.504	9.629	0.000 **
Level × Sex	72.902	3	24.301	1.042	0.373
Intervention × Sex	0.143	1	0.143	0.006	0.938
Level × Intervention × Sex	33.349	3	11.116	0.477	0.699
Error	12217.760	524	23.316		
Total	21908.150	540			
Revised total	13943.222	539			

^a^ R^2^ = 0.124 (Adjusted R^2^ = 0.099). Note: * means at the alpha 0.05 level, ** means at the alpha 0.01 level.

**Table 7 ijerph-19-13286-t007:** Independent-samples t test analysis table for the improvement of children and adolescents’ physical fitness scores by gender and level in the IG and CG.

Level/Sex	Index (Progress)	Category	N	M ± SD	T	*p*	Cohen’s d
2/Girls	Speed	IG	20	8.75 ± 3.683	2.442	0.019 *	0.782
		CG	19	5.95 ± 3.472			
	Total physical fitness score	IG	20	4.92 ± 2.379	2.368	0.023 *	0.764
		CG	19	3.30 ± 1.827			
3/Girls	Endurance	IG	22	3.91 ± 2.180	2.048	0.047 *	0.637
		CG	20	2.65 ± 1.755			
4/Boys	Strength	IG	75	16.91 ± 17.762	4.043	0.000 **	0.675
		CG	68	5.82 ± 14.997			
	Endurance	IG	75	11.56 ± 18.265	2.477	0.014 *	0.412
		CG	68	4.88 ± 13.840			
	Total physical fitness score	IG	75	4.73 ± 6.629	2.261	0.05 *	0.377
		CG	68	2.18 ± 6.895			
4/Girls	Flexibility	IG	73	3.75 ± 7.808	4.335	0.000 **	0.741
		CG	65	−1.75 ± 7.022			
	Strength	IG	73	5.71 ± 10.560	4.820	0.000 **	0.811
		CG	65	−1.49 ± 6.778			
	Total physical fitness score	IG	73	3.42 ± 4.799	3.319	0.001 **	0.565
		CG	65	0.70 ± 4.83			

Note: * means at the alpha 0.05 level, ** means at the alpha 0.01 level.

**Table 8 ijerph-19-13286-t008:** ANOVA of progress in physical fitness scores between levels and genders.

Sex	Index (Progress)	Level	N	M ± SD	F	*p*	η^2^
Boys	Lung capacity	2	20	3.60 ± 0.821	13.938	0.000 **	0.212
		3	24	5.42 ± 4.413			
		4	75	0.56 ± 1.298			
		5	40	6.88 ± 11.111			
	Strength	2	20	7.55 ± 3.220	3.384	0.020 *	0.061
		3	24	6.42 ± 2.104			
		4	75	16.91 ± 17.762			
		5	40	10.88 ± 23.322			
	Endurance	3	24	3.29 ± 2.404	4.693	0.011 *	0.065
		4	75	11.56 ± 18.265			
		5	40	4.60 ± 9.243			
Girls	Speed	2	20	8.75 ± 3.683	7.222	0.000 **	0.130
		3	22	7.77 ± 4.659			
		4	73	4.32 ± 6.794			
		5	34	2.41 ± 4.704			
	Lung capacity	2	20	5.65 ± 2.159	8.631	0.000 **	0.152
		3	22	3.64 ± 4.562			
		4	73	0.31 ± 0.116			
		5	34	4.21 ± 10.736			

Note: * means at the alpha 0.05 level, ** means at the alpha 0.01 level.

## Data Availability

The data presented in this study are available on request from the corresponding author.

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
