# Peer review of "The Effect of the KDL Active School Plan on Children and Adolescents’ Physical Fitness in China"

_ijerph, 2022, doi:10.3390/ijerph192013286_

Round 1

Reviewer 1 Report

The study exposes an important problem, however it presents important methodological and study design problems that compromise the scientific quality of the study and prevent it from being replicated.

The random selection of the sample is not explained in depth and does not fit into the structure of complete groups per classroom of the educational centers.

The lack of precise information on which intervention is applied does not allow the study to be replicated.

The validity of the measurement instruments used throughout the scientific literature is not justified.

No action is mentioned for the training of teachers who implement the intervention. There is no monitoring of the application of the program by the teaching staff. A study of the loyalty of the program is necessary to analyze the degree of implementation of this by the teaching staff. There are no guarantees that the program will meet any standards on the model put into practice.

The statistical analysis applied is not correct, a repeated measures ANOVA must be applied including the factors group, sex and age as a minimum.

On the other hand, the presentation of the work is poor in details. It does not contextualize the problem in the official documents it cites and its introduction requires a more solid justification based on international literature.

Author Response

Thank you for your valuable suggestions on this article. I have revised the contents according to your suggestions and made corresponding explanation. Please see the attachment for details. Thanks again for your guidance!

Reviewer 2 Report

Basic reporting

The authors study and analyse the changes in the physical fitness index of children and adolescents induced by the implementation of a specific and comprehensive protocol, named KDL-ASP, characterised by five domains, aimed at ensuring the performance of moderate to vigorous physical activity for at least 60 minutes per day in the school setting. The manuscript is generally well written and easy to read; a slight spell-check is required. Although the results of the study are interesting, I have some methodological and other concerns that the authors need to address before acceptance and publication. 

Abstract

·         the authors should include a background indicating to the reader the context within which the study unfolds.

·         the author should consider stating clearer the research hypothesis

·         the authors should consider explaining the meaning of the KDL abbreviations, since they’re using them for the first time in this section. 

·         the authors should improve the 'methods' section and summarise the 'conclusions' section

·         it would be appropriate to remove the 'Objective:', 'Results:', and so on

·         keywords usually should be different from that used in the main title

Introduction

The literature on the subject is sufficiently well summarised.

·         The authors should state more clearly what’s the literature gap that they’re trying to fill with their work

·         It would be appropriate to explain the abbreviation KDL, in line 26, and not afterwards since you are using it for the first time, in this section

·         Line 43-44, maybe you mean “vigorous”? I’m not sure

Methods

·         The methods section is sufficiently well described.

Validity of the findings

The results and discussion section are quite clear and organised. The parameters considered are well presented.

·         authors might consider using shorter titles for sub-sections of the results section

·         it would be appropriate to restate what the aim of your study is, in the discussion section

·         first sentence of the discussion section: are you talking about your study? If yes, why are you speaking in the plural (studies)? If not, there is a lack of citation for what you state.

·         Line 3 of the discussion section, it would be useful to indicate which of the physical fitness index of IG students are significantly different from the CG group

·         The conclusion section should be improved by briefly summarising the main findings that support the conclusions.

Author Response

(The authors gave the same response as above.)

Reviewer 3 Report

The article deals with the important and interesting issue of physical activity in children and its impact on health. The authors obtained useful results confirming the importance of physical activity for the development of children. The article should be published.

Author Response

Thank you very much for your recognition of the article! I have optimized the article in combination with other experts' suggestions. 

Round 2

Reviewer 1 Report

Although the work has improved, I consider that it still has deficiencies that must be improved.

The introduction is poor in the contextualization of the problem based on the current international literature with the absence of consistent theoretical frameworks that support the work developed. Although a theoretical structure of the study (KDL-ASP) is included, the design of the study does not respond to this structure since the teaching staff, the environment, the class and the playground are not part of this study, since they are variables that are not studied. The arguments used are based on obvious evidence, if something is worked on, it improves. The problem of the practice of physical activity is in the acquisition of knowledge and habits that are maintained over time without the need for the presence of an agent, be it an educator or a health professional. I consider that the approach of the work is not original, nor does it contain processes and information that may be relevant in the current scientific context. Still, the study is well thought out.

On line 3 of page 4 you have to check “invovled”

The question of how the control and experimental groups were assigned to the student level groupings has not been answered. Were they randomly assigned or for convenience? How was it made?

The validity of the information in research studies is a fundamental criterion. The tests and bibliographic references that support them, the instruments used, the procedures followed for their use, the experimental situation in which the data are collected, the inclusion of analysis of the reliability of the measures of those responsible for collecting the data, the error measures between judges... these, among others, are validity factors that are not addressed in the study.

Reviewing the dates of the intervention it is striking that the period indicated by the researchers coincides with the period of confinement due to the pandemic. How was confinement managed in this process?

Regarding how the professionals who implement the intervention program are trained, the answer does not clear up the doubts since the information provided is not very precise. How many professionals participated? What materials did they receive? What kind of training did the professionals receive? How long have they been in training? How were they evaluated to ensure that they met the fundamental principles of the program? How was the fidelity of the implementation of the intervention evaluated in each of the groups in which it was intervened?...

In table 5, level 1 is not displayed.

The analysis of the data should explain more precisely the type of multivariate analysis applied and how it is applied.

The methodology explains that data is collected to calculate the body mass index. If data is collected on this variable, why is it not used?

Including the effect size in the contrasts presented in Tables 7 and 8 would improve the quality of the study data.

Some of the values in Figure 4 are not visible.

The discussion is rich and appropriate as well as the conclusions of the study

Author Response

Thank you for your careful and rigorous reading, as well as your questions and suggestions on this article, so as to provide good reference for our further research. I have revised and explained each item according to your comments in the attachment. I hope to get your approval. Best wishes!
